# Parameterization in the Analysis of Changes in the Rural Landscape on the Example of Agritourism Farms in Kłodzko District (Poland)

**Anna Bocheńska-Skałecka** [1],*[ORCID], **Maria Ostrowska-Dudys** [2], **Edward Hutnik** [2] **and Wojciech Jakubowski** [3]

1  Department of Landscape Architecture, Wroclaw University of Environmental and Life Sciences, 50-357 Wroclaw, Poland
2  Department of Civil Engineering, The Faculty of Environmental Engineering and Geodesy, Wroclaw University of Environmental and Life Sciences, 50-357 Wroclaw, Poland; maria.ostrowska-dudys@upwr.edu.pl (M.O.-D.); edward.hutnik@upwr.edu.pl (E.H.)
3  Department of Applied Mathematics, The Faculty of Environmental Engineering and Geodesy, Wroclaw University of Environmental and Life Sciences, 50-357 Wroclaw, Poland; wojciech.jakubowski@upwr.edu.pl
*  Correspondence: anna.bochenska-skalecka@upwr.edu.pl

**Abstract:** The European Landscape Convention (2006) indicates that landscape conservation is as important as the protection of the overall environment. Although the boundaries between urban and rural areas in many countries are blurring, the rural landscape is still perceived as a valuable landscape artefact. Traditional rural landscapes have undergone significant transformations over the past few decades. The authors attempt to analyze factors causing apparent changes in the rural landscape, based on the example of agritourism farms in Kłodzko District, Lower Silesia. The changes taking place in Poland after 1989 resulted in reduced profitability of agricultural production. This was why small farms stopped using land for agricultural production. Agritourism has become one of the forms of business activity. Therefore, it became necessary to adapt farms to a new function. The 37 agritourism farms registered in rural and rural-urban municipalities of Kłodzko District have been randomly selected for the survey. The research has shown the extent of changes related to the transformation of agricultural farms into agritourism ones. Six areas (categories) where changes took place have been identified based on the analysis of collected data. The authors have included the collected data in the parameterization of surveyed agritourism farms, taking into account: the condition of the agricultural farm before introducing its new role (0) and the present condition, with an agritourism function (1). The complete linkage clustering (the maximum distance) known as cluster analysis was used to examine the variables in terms of farm change. The aim was to select outstanding units from the research sample for further research as case studies.

**Keywords:** rural landscape; agritourism; cluster analysis/ complete linkage clustering; Kłodzko district

## 1. Introduction

Poland has been a member of the European Union for over fifteen years. This gave the Polish government the opportunity to benefit from EU programs and projects supporting the development of various areas of the economy including agriculture. The first agritourism farms started to be established benefiting from financial support for EU countries [1–3]. As a result, farmers began to adapt rural homesteads to a new function—tourism [3,4]. The beginning of the 21st century was marked by a noticeable demand for 'farmer's accommodation and a sudden increase in the number of farms that offered holidays in a reconstructed homestead or newly built facilities. Unfortunately, the protection of natural and landscape values has been forgotten [5], in the spontaneous modernization process of buildings. Over time, following the example of the European thematic villages, some farmers have chosen a guiding direction for the development of

their homestead, subordinating all modernization measures to it (e.g., painting the buildings in one color), giving a farm its own name referring to the characteristic features of the area or a regional product [6]. Currently, newly established agritourism facilities address their offer to a specific customer and seek to adapt it to the anticipated needs of future visitors. Many farmers have left livestock production or have kept agricultural production to a minimum, which has completely changed the space of the homestead [7]. This often results in agritourism farms spaces which have no rural character and do not retain the features of regional buildings [8,9].

Tourism in agritourism farms is identified with rural tourism, while its origins date back to the 19th century. At that time, holiday villages and spending leisure time at country estates were popular [9,10]. Rural tourism includes agritourism as well as recreation in real rural areas related to nature, hiking and health, landscape, cultural and ethnic tourism with particular emphasis on the values and resources of villages [11,12]. Return to a traditional form of community life in rural areas and contact with nature takes place through rural tourism with the distinction of agritourism [13,14]. Agritourism is an essential activity for developing rural areas in terms of rural cultural tourism to which it belongs. At the same time, it is a place to learn about tradition, customs, culture and what remains after the former village [11,15].

Recreation is one of the most important cultural ecosystem services in the European context [15,16]. Many studies highlight the importance of the intangible benefits provided by ecosystems, especially cultural landscapes, which are also shaped by human interaction with nature [16]. Various research methods were used to analyze the impact of agritourism farms on the rural landscape, which enabled the authors to visualize the change in the settlement unit that occurred due to the change in its function [17].

## 2. Literature Review

### 2.1. The Agritourism Farm in the Context of the Multi-Functionality of Rural Areas

According to Polish legal acts, rural areas are defined as areas located outside the administrative borders of cities (i.e., areas of rural municipalities or parts of rural-urban municipalities) [18]. The rural area is a human-made space characterized by low population density, scattered settlements and extensive land use [11]. The definitions found in the literature underline the importance of the following features in rural areas [19,20]:

- Specific open landscape;
- Low population density;
- Predominance of people involved in farming, forestry and tourism;
- Traditional lifestyle (close to nature) and cultivation of customs;
- Agricultural and forestry use of land;
- Sparse buildings and dispersed settlement;
- Inhabitants' feeling of living in the countryside.

According to Article 2 of the Act on official names of places and physiographical objects, a village is a settlement unit with compact buildings and existing agricultural functions, related service functions or tourist functions, which does not have urban rights or city status (the Act of 2003 on official names of places and physiographical objects). Until recently, rural areas were dominated by agricultural land use and the associated occupational structure of the population and the relationship between work and residence. Changes in the functioning of villages have resulted in their transformation into a model of a diverse village [21,22]. The European Union sees great potential in rural areas, as they account for as much as 80% of its territory. Rural areas are characterized by a variety of cultures, natural resources and historical monuments. At the same time, they are attractive places for social life, residence and tourism, including rest, recreation, entertainment, gastronomy, communing with culture and nature [16,23]. Polish rural areas are inhabited by around 38% of the country's population, while they cover over 90% of the country's area. They are therefore of great importance in economic, social and environmental terms [24–27]. The open landscape of rural areas is important in terms of tourism activities, especially in

the case of a variety of mountainous regions [28–32]. Noted that these aspects contribute often to considerations of multifunctional rural development and landscape transformation at the local level [33–39].

### 2.2. Methods of Valuating the Rural Landscape

Human activities in the natural environment, including rural areas, use research methods of technical sciences to solve technical, biological and structural problems [40–42]. Their use enables the diagnosis and description of the technical condition and the conditions for the implementation of the adopted method of procedure with the diagnosis of the effects resulting from the measures taken and the implementation of technical and natural undertakings affecting the sustainable development of rural areas [43].

The source literature distinguishes three main groups of methods for landscape assessment and valuation [40–43]:

- Assessment of the natural value of individual elements;
- Aesthetic and visual assessment of valuable landscapes;
- Valuation of landscape for a specific purpose.

In addition to the classification according to the element to be assessed, the methods can be divided according to the way the information is obtained (field, chamber and mixed methods) [43–45]. In terms of the scope for using evaluation and valorization methods, partial and comprehensive methods can be distinguished [42–45]. The UK Landscape Character Assessment (LCA) method of landscape character assessment (two stages: characterization and assessment) [43–45] is used in rural landscape planning and management. It is suitable for national, regional and local use [45]. It is used in many European countries for landscape planning [43–45]. This is complemented by Historic Landscape Character (HLC), which assesses the retained historic qualities of the landscape [44,45]. The method aims to characterize and interpret the visible historical elements in the landscape of area given area. Quality of Life Capital (QLC) complements LCA as it enables the study of people's quality of life in a given environment [46,47]. The method is also used in terms of studying human perception of the landscape. There are also tools to enhance rural landscapes of the Village Design Statement type, which provide a set of design guidelines for rural settlements [47]. The use of such tools helps to integrate redeveloped existing buildings and new buildings into the rural landscape [47].

Scenic Beauty Estimation (SBE) is a method used to assess the aesthetic value of landscapes, originally developed for forest areas [48]. The method involves taking 10–15 photographs of each unit and having them assessed by a group of observers from different social backgrounds. The final grade is the result of the mean with the standard deviation applied [48,49]. The WNET method of Cymerman and Koc, on the other hand, differs from the previous one in that only natural values are analyzed as those that increase the value of the landscape and its ecological usefulness [50]. Another method based on the criterion of aesthetic value is Bajerowski's matrix method, which assumes the use of data from topographic and land register maps to identify spatial features [50,51]. The following methods are used to assess the degree of landscape transformation in terms of human activity:

- Assessment of the proportion of straight line in a view or panorama [41,50,51];
- Recording Wejchert's "impression curve" [41,49–51];
- Assessing the proportion of natural, semi-natural and anthropogenic land cover forms [40,41,49,51];
- Assessing the proportion of natural, semi-natural and anthropogenic landscape boundaries [49,51].

Landscape can also be studied depending on the level of hierarchical division of landscape space, starting from the most general level: landscape, architectural and landscape zone (strefa architektoniczno-krajobrazowa, SAK), architectural and landscape unit (jednostka architektoniczno-krajobrazowa, JARK), architectural and landscape interior (wnętrze

architektoniczno-krajobrazowe, WAK), [41,49–51]. A multi-criteria valorization assessment method, in which several groups of criteria are assessed, is also used in rural landscape valorization [52]. The study area is subjected to a series of assessments:

- Natural valorization;
- Cultural valorization;
- Agricultural production space valorization;
- Landscape aesthetic assessment;
- Valorization for different forms of recreation;
- Assessment of the degree of anthropogenic transformation of the environment;
- Assessment of conflicts and conflict areas.

Research models have also been developed to assist in the valorization of cultural landscapes. For example, Raszeja [26] developed the research model Biography-Structure-Image as a scheme for dealing with the process of perception, identification and interpretation of cultural landscapes. The author assumed that the cultural rural landscape is a unique spatial structure with encoded historical processes, geographical relations, cultural symbols and meanings. The concepts that define the model (biography, structure and image) are learned in the process of understanding the landscape. A similar scheme was used to learn about the cultural landscape of the rural area analyzed in this article.

### 3. Study Area

The district of Kłodzko is located in the south-western part of Poland (Figure 1), surrounded by a neighboring country, the Czech Republic, on three sides. It is situated in the southern part of Lower Silesia Province, in the subregion of Wałbrzych and borders the Czech Republic. It has borders with the district of Wałbrzych to the north, the district of Dzierżoniów to the west and the district of Ząbkowice to the north-east. The district of Kłodzko is the largest one in Lower Silesia Province with an area of 1642 km$^2$ [53] and one of the largest in Poland (25th place in terms of area in the country [54]).

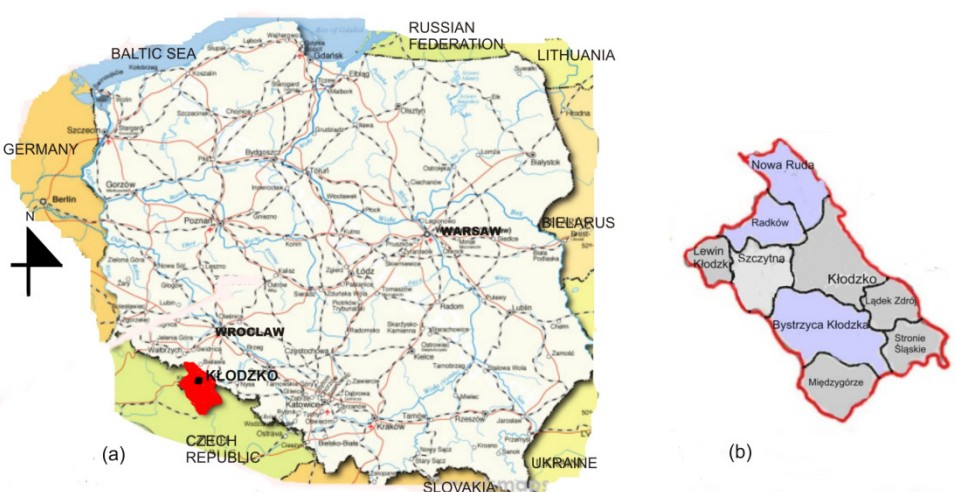

**Figure 1.** Location of the district of Kłodzko—research area (**a**) on a national scale, in Poland (**b**) research area—the Kłodzko district—division into municipalities.

In the center is a basin—Kotlina Kłodzka (the Kłodzko Basin), which is surrounded the Bystrzyckie, Orlickie and Stolowe Mountains in the west, the Sowie and Bardzkie Mountains in the north, the Śnieżnik Massif in the east, the Bialskie and the Złote Mountains. The border with the Czech Republic runs along the tops of these mountains, dividing them into northern slopes belonging to Poland and southern slopes belonging to the Czech Republic. Kłodzko District is the largest in Lower Silesia Province and ranks second in terms of population, with 162,465 people, i.e., almost 100 people per square kilometer (provincial average). This accounts for about 5.5% of the population in Lower Silesia

Province. Most people live in urban areas (64%). Excluding urban municipalities, the population in rural-urban municipalities exceeds 90,000, of which 58,070 people live in rural areas representing 64% of the total in these areas. It follows that the population density is almost a half lower than the province average. That means rural areas are not densely populated (59 persons per km$^2$) [55]. The strategy for agriculture and rural development (2000) identifies five regions of functional rural areas in Lower Silesia Province. Kłodzko District is located in the 3rd Region—industrial, recreational and tourist region, which includes the subprovince of the Sudetes and Przedgórze Sudeckie (the Sudetian Foothills) [54,55]. In addition, it is part of Sudecki Obszar Integracji (the Sudeten Integration Area) (Area C). Area C is characterized by some development barriers, which originate from the depopulation process and the high unemployment rate. The Sudeten Integration Area is featured by unique natural and landscape values (e.g., two national parks), as well as high tourism and spa potential that has not been fully exploited [55,56]. The development of the area is dependent on cross-border cooperation with the Czech Republic (the Development Strategy for Lower Silesia Province for 2013, 2020). The Development Strategy for Lower Silesia Province (2020) defines Intervention Areas, where phenomena or processes related to spatial conflicts occur. Kłodzko Region is also included in these areas. There are 8790 farms (including 8756 individual farms) in the district of Kłodzko. 6982 farms were involved in agricultural activities [57]. On the other hand, 1572 farms in Kłodzko District declared income also from non-agricultural activities (one of them was agritourism) [57]. The study of the delimitation of rural agritourism space in Poland [55–57] shows that Kłodzko District stands out as an attractive agritourism region in the Sudetes and Przedgórze Sudeckie.

## 4. Methodological Approach

### 4.1. Research Trial

Depending on the adopted source of figures, the number of agritourism farms varies in rural and rural-urban municipalities. To determine the number of farms for the statistical survey, they were randomly selected based on the number in the municipal registers. That is, sixty agritourism farms were chosen at random. The selected farms were visited during fieldwork, five of which proved to be non-active, four were found to be inactive despite being listed in the municipal register. Thirteen farm owners refused to participate in the survey. After a walkover survey, thirty-seven agritourism farms registered in rural and rural-urban municipalities of Kłodzko District (15% of the total number of farms) were selected for further research. The designated agritourism farms are located in seven municipalities of Kłodzko District: Kłodzko, Stronie Śląskie, Radków, Nowa Ruda, Szczytna, Bystrzyca Kłodzka and Międzylesie. Only two agritourism farms are registered in the municipality of Szczytna, so both farms have been surveyed in this case (Figure 2).

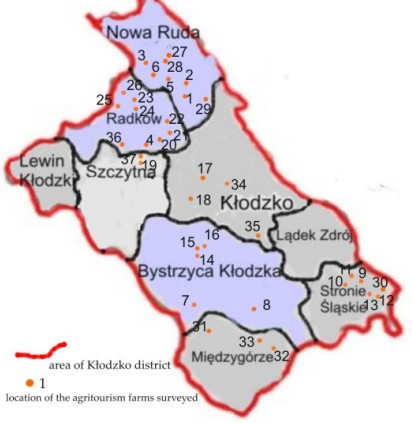

1…37   identification numbers of the surveyed agritourisms

**Figure 2.** Location of the agritourism farms finally surveyed in Kłodzko district.

### 4.2. Field Research

The literature review and field research allowed the technical state of agritourism farms to be determined. The collected and compiled data were also used to present the architectural and landscape features of the surveyed agritourism farms. This phase of the research aimed to identify opportunities for their further development. The data were compiled based on the basic information received during an interview with an owner and then compared against each other. The surveyed farms were randomly selected based on the registration of agritourism activities maintained by the municipalities. The majority of agritourism farmers were of working age (over 70%). As a reason for the start-up of agri-tourism activities, they reported additional income and having vacant premises and buildings. On the other hand, people at the post-productive age have also declared the use of "free spaces" and the need for contact with people. Most of the farmers started their agritourism activities in the 2000–2010 period, which was affected by Poland's accession to the European Union and the possibility of obtaining funding for such activities. Most of the farms were dominated by old buildings. Land with such buildings was usually inherited from parents. Several cases are inherited farms with old buildings on which new facilities for agritourism services have been built. In very few cases, the land was purchased for building a new agritourism farm on it. Those wishing to make a living from agritourism were most often looking for old farms to renovate. Due to the need to adapt the farm to its new function, 87.5% of the owners have carried out renovations and upgrades. According to interviews conducted with owners, 75% of them are still investing in the further development of their farm and see the need for changes. Development plans are usually to extend the offer and to adapt social facilities and space to provide accommodation. This applies in particular to farmers who have used loans or funding from the European Union to run their agritourism activities. More than 55% of the farmers were involved in mixed production (crop and livestock) before starting their business. Half of them abandoned animal production, retaining only crop production after introducing the new function. However, there were also farms whose owners started agricultural production when they started their agritourism activities. The majority of the analyzed farms (41%) did not exceed an area of 2 ha. However, farms of up to 20 ha constitute 39%. Both groups were in the submountain areas and the soil quality class was low. Only 20% of the farms covered an area of over 20 ha and included agricultural land in river valleys. In the majority of farms (57.5%), in addition to residential buildings, there were livestock and storage facilities. Existing storage and livestock buildings have often been adapted as guest accommodation. As a result, new buildings have been erected on the farm, i.e., garages, workshops or buildings for special technical or farming purposes. Existing farm buildings of large cubic capacity were often adapted to new functions related to agritourism services (e.g., recreation room, dining room, and sauna).

### 4.3. The Extent of Changes in the Farm When Changing Its Function from an Agricultural to Agritourism

The research carried out on a sample of thirty-seven agritourism farms identified the extent of changes that were made when a farm was transformed into an agritourism farm. This involved a change in the function of the space or adding a tourist function—a separate space for guests. Six areas where modifications took place have been identified based on the analysis of collected data (Figure 3). It indicates that most of the remodeling took place within the residential building (93% of the surveyed farms), of which 54% of the renovations concerned the installation or modernization of central heating. 51% of farm owners declared that they renovated and reconstructed a roof (roof covering, change of roof pitch or transforming an unused attic into a habitable one—lighting). The changes involved raising a roof to obtain an additional floor, installing additional roof windows and dormers, insulating a façade, changing the color scheme and finishing materials of the building façade, replacing sanitary and heating installations, draining the building, constructing a new building or extending the existing one. 62.5% of farm owners

modified the farm building. 72% of modifications (18 out of 25 modifications) concerned a transformation from farming to residential function(adaptation). The transformation involved renovating the façade and roofing and extending the building. The function of more than half of the buildings (55%) has changed from farming to residential. In 55% of cases, the changes concerned small architecture and greenery, of which 100% were the construction of new objects or elements: fences, seats, benches, tables, barbecue areas. Modifications to the greenery included new planting and restoration of existing vegetation to a lesser extent. In about 40% of farms, the existing vegetation has been maintained. Tree rows, hedges, lawns, front gardens, vegetable gardens and house orchards were also replenished. More modifications have been carried out in the recreational space area (62%) than in the green space area (playgrounds have been created). In 75% of the surveyed farms, the technical infrastructure has been alternated. Parking spaces and car parks have been marked out on the farm, surfaces have been paved and areas has been fenced. Roads and maneuvering areas on the farm have been hardened or paved. Parking area lighting has also been installed in 30% of the farms. Modifications concerning: buildings, small architecture structures, technical infrastructure and greenery were taken into consideration for further research.

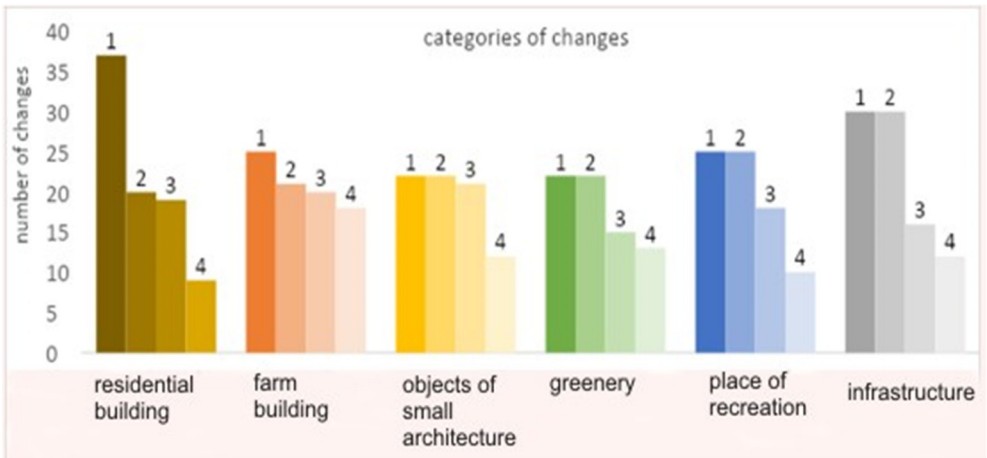

**Figure 3.** Number and distribution of changes within surveyed farms by category. Legend: residential building: 1—number of changes made, 2—installations (heating system), 3—roof renovation, 4—new building). farm building: 1—number of changes made, 2—building extension, 3—renovations, 4—building conversion; objects of small architecture: 1—number of changes made, 2—new construction, 3—renovation of old, 4—reconstruction. greenery: 1—number of changes made, 2—new planting, 3—maintenance of existing, 4—greenery restoration. place of recreation: 1—number of changes made, 2—grill/fireplace, 3—playground; 4—renovation of existing; infrastructure: 1—number of changes made, 2—demarcation of a car park, 3—surfacing the car park, 4—installation of lighting.

*4.4. Assessment Categories for Agritourism Farms*

Based on the literature, to determine spatial changes in agritourism farms, general features were extracted. They could give a view of the situation before the agritourism activity and the current state. On the other hand, compilation of renovations and changes made to the farms has been used to define the categories for assessing farms in terms of the presence of specific characteristic. It was observed that the greatest changes were to the residential building. Accordingly, the first category includes assessing the regional (local) characteristics of residential development. In the first category—the homogeneity of the regional style—the following features were distinguished: the color and form of the roof (roof slope inclination) and the body and covering of the building façade. These are the features that are most visible when observing buildings on a landscape scale [7–9,58,59]. The second category—the quality of the panoramas—determines the location of the farm in space: whether the site is in a prominent place in the field or whether it

is located in the vicinity of similar buildings [41,43–45,49,50]. Another category concerned the impact of the farm on the natural environment—the third category—environmental impact. Features such as the use of natural, non-toxic, local building materials have been highlighted [9,34–38,43,60]. The fourth category—local law—assesses the adjustment of the analyzed farms to the applicable local laws [1,21,37,39,40,42]. Another fifth category—technical condition—determines the wear of the surface materials of the farm buildings and the technical infrastructure itself [3–5,53,61,62]. The sixth and last category—agritourism space: greenery and details—assesses the condition (quality) of the agritourism space where guests stay. This category relates to determining the agricultural past of the farm and the preservation of its rural character [4,13,14,28,53,54,57,59,61,63,64].

### 4.5. Assessment Data Interpretation Method

The collected data were entered into parameterization mechanisms. According to these mechanisms, the characteristics of each test object (agritourism farm) were determined. Each category of parameterization was given a set of characteristics to describe it (Table 1).

**Table 1.** Features that describe the categories of parameterization.

| Category | Feature Group | Feature (Rating) |
|---|---|---|
| Category I The homogeneity of the regional style | 1.Roof angle: | 55° ≥ x ≥ 45° (+2)FAR; 55° ≥ x ≥ 45°, dormer windows (+1)FAR; 55° ≥ x ≥ 45°, roof windows (0); 30° ≥ x or flat roof (−1); 30° ≥ x or flat roof, dormer windows (−2). |
| | 2. Roof material: | shingle/slate (+2)FAR; ceramic, concrete, metal roofing tiles, roofing felt (imitating shingle/slate) (+1)FAR; ceramic, concrete, metal roofing tiles, roofing felt (not imitating shingle or slate) (0); papa / thatch (−1); eternite (−2). |
| | 3. Roof color: | dark brown, graphite (+2)FAR; brown (+1); light brown/black (0); red/orange (−1); blue/green/other (−2). |
| | 4. Roof shape: | pitched 2/3 of the height of the building (+2)FAR; pitched higher than the walls (+1); pitched symmetrical (0); pitched, broken (asymmetrical) (−1); multi-hipped/flat (−2). |
| | 5. Building material: | first floor-wooden, ground floor-brick (+2)FAR; first floor painted, ground floor plastered (+1); white plastered (0); plaster other than white or panels (−1); other materials glass/slate/mosaic (−2). |
| | 6. Shape residential building: | permissible dimensions: h = approx. 10m (h farmhouse = 8, h garage = 6) (+2)FAR; dimensions in the ratio 1 wall: 2 roof (+1)FAR; dimensions in the ratio length: width 2:1 (0); dimensions in unacceptable proportions (−1); exceeding the permissible dimensions (−2). |
| | 7.Residential building construction: | wooden (residential function)-brick (livestock f.) (+2)FAR; brick (residential function)—wood (livestock function) (+1); mixed (function does not depend on construction) (0); brick built as a whole (−1); entirely wooden (residential and utilitarian) (−2). |
| | 8.Building details: | eaves-wooden carved(+2)FAR; arcades, balcony galleries/boarded gables (+1)FAR; white window and door bands(0);window bands other than white (−1); windows without bands, various shapes (−2). |
| | 9. Types of buildings in the yard: | combined building with residential and livestock functions (+2)FAR; composite building with a residential function (+1); non-segregated buildings (0); a residential building and a non-agricultural building (e.g., garage, shed) (−1); only residential building (−2). |

<div align="center">

**Table 1.** *Cont.*

</div>

| Category | Feature Group | Feature (Rating) |
|---|---|---|
| Category II The quality of the panoramas | 1. Transparency of space: | multiplanarity of the view (+2); no infrastructure (open view) (+1); vertical point infrastructure (0); overhead linear infrastructure (−1); infrastructure obstructing the view (−2). |
| | 2. Landscape elements of the interior: | background, walls, floor, partitions—visible, (more than 4 plans) (+2); background, walls, floor—visible, partitions on walls (+1); background, walls, floor—visible (0); floor—visible, background—visible (−1); floor—visible, background—covered (−2). |
| | 3. The ratio of greenery to buildings: | greenery higher than buildings(+2); greenery and buildings of equal height (+1); buildings higher than greenery (0); high buildings, low greenery (−1); high buildings, no greenery (−2). |
| | 4. Accompanying greenery characteristic of a traditional village: | solitary/large trees/shrine/cross (+2); avenue/row (+1); low vegetation, e.g., crops, meadow (0); low grassy vegetation (−1); no greenery (−2). |
| | 5. Location of agritourism in relation to other buildings in the village: | compact—objects on one side (+2); compact—buildings situated on both sides (+1); solitary (0); loosely built-up area (−1); in the second line of development (−2). |
| | 6. Landscape values: | on a viewing axis or vantage point (+2); the exposition foreground (+1); panorama components present (0); no panorama components (−1); no exposure (−2). |
| Category III Environmental impact | 1. Source of energy: | heat pump, solar collectors (+2); co-generation furnace (+1); mains heating (0); solid fuel cooker (−1); fireplace (as the only one for heating) (−2). |
| | 2. Eco-friendliness of the materials used: | environmental protection materials (e.g., insulation, joinery) (+2); filtering materials car park with protected floor (+1); lack of environmental protection materials (car park with substrate, no insulation, standard window joinery) (0); lack of environmental protection materials (e.g., car park surface, poor quality window joinery, slurry boards, cavities in plaster "thermal bridges") (−1); harmful material, e.g., asbestos (−2). |
| | 3. Waste management: | compost, segregation, rainwater harvesting (+2); infiltration plant (+1); sewage treatment plant, waste separation (0); septic tank, no waste separation (−1); no sealed tank/landfill (−2). |
| | 4. Farm status: | certified organic (+2); organic without certification (e.g., transitional period) (+1); conventional farm (0); intensive or monoculture holding (−1); large-scale enterprise (−2). |
| | 5. Land infrastructure: | lack of visible elements. technical infrastructure (+2); concealed infrastructure elements (+1); wells (0); poles, wells (−1); trafostacja (−2). |
| Category IV Local law | 1.Use of the area.: | as intended and for an additional purpose (development of activities) (+2); as intended and used for additional purpose (+1); as intended (0); compatible with the local plan, but location of incompatible functions (−1); inconsistent with the local plan (−2). |
| | 2. Building conditions (administrative decision, guidelines: | massing and floors in accordance with the guidelines (+2); development with a number of floors according to the guidelines (+1); development consistent with the function (0); development not in line with guidelines (solid) (−1); buildings with inconsistent number of floors, poor distribution of window surfaces, façade materials with guidelines (−2). |
| | 3. Land cover (degree of development of biologically active land): | additional biologically active surface (e.g., on a wall or roof) (+2); less than 30% of the plot area built up or permeable surfaces (+1); built-up area of approx. 30% of the plot area (0); developed area in more than 30% or impermeable area (−1); built-up area of more than 50% of the plot (−2). |
| | 4. Recreational suitability by way of use: | recreation on landscaped grounds (linked to the surroundings) (+2); services related to recreation (cultural, sports, tourism—ruins) (+1); elements supporting recreation (garden, small architecture, forest) (0); objects indifferent to recreation (housing, wasteland, agricultural land, shop) (–1); objects interfering with recreation (industry, warehouses, transport, railway) (−2). |
| | 5. Recreational suitability by way of use in accordance with the local land development plan: | permanent (investment in agro-tourism) (+2); residential, amenity greenery (barbecue, playground) (+1); temporary or mixed (including those not in conformity with the plan) (0); incompatible with the function (−1); undeveloped (no purpose) (−2). |

**Table 1.** *Cont.*

| Category | Feature Group | Feature (Rating) |
|---|---|---|
| Category V Technical condition | 1. Technical condition of the roof: | very good (new/renovated) (+2); good (+1); average (requires refurbishment/maintenance) (0); sufficient (requires renovation) (−1); bad (likely to collapse) (−2). |
| | 2. Technical condition of the facade: | very good (new/renovated) (+2); good (+1); average (requires refurbishment/maintenance) (0); sufficient (requires renovation) (−1); bad (likely to collapse) (−2). |
| | 3. Technical condition of the farm building and another: | very good (new/renovated) (+2); good (+1); average (requires refurbishment/maintenance) (0); sufficient (requires renovation) (−1); bad (likely to collapse) (−2). |
| | 4. Technical condition of the infrastructure: | very good (new/renovated) (+2); good (+1); average (requires refurbishment/maintenance) (0); sufficient (requires renovation) (−1); bad (likely to collapse) (−2). |
| Category VI Agritourism space: greenery and details | 1. Functional zones: | residential, manufacturing, recreational, accessory (e.g., educational, therapeutic) (+2); residential, production, recreational (+1); residential, production (0); residential, utilitarian (−1); residential (−2). |
| | 2. The presence of greenery: | ornamental, recreational, amenity garden (+2); ornamental, amenity garden (+1); ornamental garden (0); no garden (only green areas) (−1); no garden and other greenery (−2). |
| | 3. Garden style—vegetation: | in keeping with the region and style, rural (+2); in keeping with the region or style (+1) rural; not compatible with the style but local (rural) (0); not compatible with the style but region (−1); not compatible with the region and not rural (−2). |
| | 4. Garden style—small architecture and details: | regional materials (+2); materials imitating regional character (+1); neutral materials (wood/stone) (0); material not found in the region (−1); distinctive material (plastic/metal) (−2). |
| | 5. Garden style -nature of coverage: | preserved old composition (numerous plantings) (+2); retained old planting (single) (+1); species compatible with the former layout (0); species incompatible with the former layout (−1); layout incompatible with former layout (modern style, urban development) (−2). |
| | 6. Garden style -coverage variety: | unique natural values (+2); vertical diversity (+1); horizontal diversity (0); no vertical diversity (−1); no horizontal diversity (−2). |
| | 7. Visible cultural heritage: | numerous items of cultural heritage (machinery and former agricultural equipment) (+2); isolated items of cultural heritage (pots and pans, barrels, etc.) (+1); modern agricultural equipment(0);non-agricultural equipment (−1); scrap—non-agricultural items (−2). |
| | 8. Accompanying buildings in the farm: | cultural heritage sites (windmills, watermills, chapels, etc.) (+2); traditional agricultural sites (barns, stables, etc.) (+1); modern agricultural buildings (silos, garages, sheds, shelters, etc.) (0); large-scale buildings (−1); other objects not related to agriculture (wind turbine, mast, chimney, etc.) (−2). |
| | 9. Advertising and informational objects in the farm: | billboards and decoration made of natural materials (+2); advertising from natural materials (+1); no advertising or information elements (0); lettering, graffiti, neon signs, city-like information signs (−1); large advertising, billboards, etc. (−2). |

Legend: FAR, feature of the regional architecture.

Each attribute contains a description with a score ranging from "+2" to "−2". A score of "0" is assigned to basic properties that are encountered most frequently and do not require special features from the object under examination. A score of "+1" has a description of an increased trait, while "+2" represents the highest degree of a trait occurring in the surveyed object. Accordingly, an object having a slight negative deviation of a given characteristic gets a "−1" or "−2" score (Table 2).

In total, each farm could obtain a maximum of (9 + 6 + 5 + 5 + 4 + 9) = 38 × 2 points = 76 plus points in the six categories. If the sum of all points is "0", it means that the agritourism farm meets the minimum requirements set by the parameterization. For example,

under category I—homogeneity of the regional style—(nine characteristics surveyed), the farm could obtain a maximum of 18 positive points (9 × 2 points). Differences in scores between sites have been determined using the maximum difference method. The obtained deviations (+, 0, −) have been assigned colors, respectively, for better readability of the results (Figure 4).

**Table 2.** Summary of grade deviation ranges from the mean.

| Deviation Range | Color | Evaluation |
|---|---|---|
| from (+20) to (+10)<br>high | green | plus rating |
| from (+9) to (+1)<br>average | yellow | average rating |
| 0<br>lack of | orange | 0 |
| from (−1) to (−x)<br>low | red | minus rating |

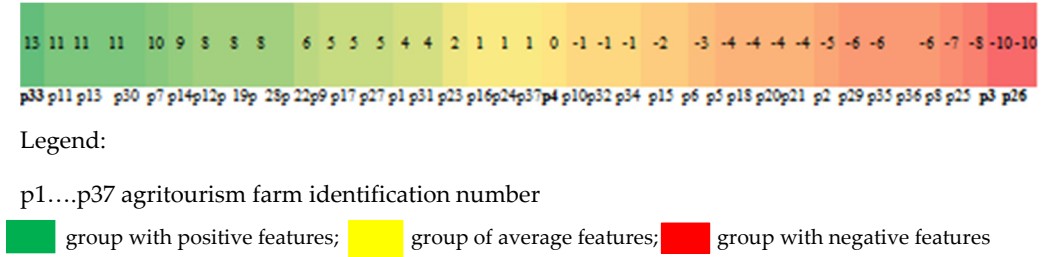

Legend:

p1….p37 agritourism farm identification number

group with positive features;    group of average features;    group with negative features

**Figure 4.** Division of ratings of agritourism farms into groups with similar values for the category I The homogeneity of the regional style.

Within each category, a score was given to each research unit (farm) and then added up. The different categories were analyzed for differences in scores between test subjects. This combination made it possible to divide the studied units into groups with positive, average and negative characteristics. Within a given group, it was possible to carry out an analysis comparing farms with each other and to identify those which had similar characteristics. By summing up the categories, it was possible to distinguish farms with similar characteristics and identify objects for further case study research. Agritourism farms were subjected to a parameterization. The result was an individual category value and an aggregate score (Figure 5).

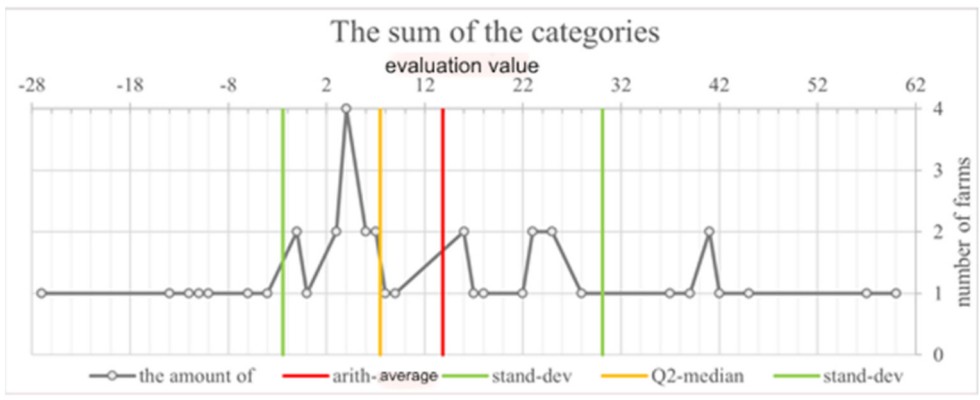

**Figure 5.** The graph illustrate summary rating of all categories.

Using the parametric values, statistical parameters were calculated, specifying arithmetic average, median, dominant, standard deviation and coefficient of collective variation (deviation %) (Table 3).

**Table 3.** List of statistical parameters of the parametric evaluation category of agritourism farms.

| Parameter | Category I | Category II | Category III | Category IV | Category V | Category VI | Sum Categories |
|---|---|---|---|---|---|---|---|
| X (arithmetic average) | 1.025 | 3.075 | 0.775 | 3.35 | 3.65 | 1.975 | 13.85 |
| $Q^2$ (median) | 0.5 | 3 | 1 | 3 | 4 | 1 | 7.5 |
| Dx (dominant) | none | 2 | 1 | 2 | 4 | none | 2 |
| Sx (standard deviation) | 5.53 | 2.88 | 2.06 | 2.19 | 1.61 | 6.72 | 16.29 |
| V(x) (coefficient of variation of collective | 539% | 94% | 266% | 65% | 44% | 340% | 118% |

### 4.6. Cluster Analysis Furthest Neighbour (Complete Linkage), the Maximum Distance

The parametric values of agritourism farms were described statistically then cluster analysis was carried out. This method involves grouping research units into clusters with similar characteristics. The grouping of objects is intended to create clusters between objects in the same group with the highest possible degree of connection and objects from other groups with the lowest possible degree of connection (StatSoft, Inc., Tulsa, OK, USA 2006) [65]. Relating objects to each other is carried out by specifying the distance between them. The complete linkage method (i.e., the furthest neighbor method has been used). The greatest distance was measured. The distance measure was obtained by processing the values of parameterization as well as its categories. The distance was calculated using the formula (1). In Equation (1) *k* represents *assessment category*, *c* represents *assessment features*; *n-nb* is the feature; *p-nb* is the farm.

$$distance(x,y) = \sum_1^k \left( \sum_{i=1}^{p-1} \left( \left| c'_{p,k} - c'_{p+1,k} \right| + \left| c''_{p,k} - c''_{p+1,k} \right| + \ldots + \left| c^n_{p,k} - c^n_{p+1,k} \right| \right) \right) \quad (1)$$

Maximal distance is a measure between clusters determined by the largest distance between any two objects from different clusters ("furthest neighbors") (StatSoft, 2006) [65,66]. A graphical way of analyzing a typological set of units is called a dendrite [65,66]. A dendrite maps the structure of a set of objects. It is the arrangement of objects of similar nature closest to each other. The distance on the dendrite indicates mutual similarity. The greater the distance, the more it allows to observe subgroups and typological groups of objects with as many common features as possible. The greater the distance between groups and objects, the more typologically different the features are. In this way, a tree diagram of the elements of the analyzed set can be created based on clusters. A tree diagram which presents farms with similar characteristics, was created (Figure A1). The tree structure ordered the farms in a distance matrix. Groups with distinctive characteristics were then identified. The matrix transformed in this way then shows a dendrite (a graph) that indicates farms that are close to each other due to similarity of characteristics. The computer program GNU Octave was used to group farms with similar characteristics, i.e., to transform the distance matrix (Figure A2).

*4.7. Methodology for Analysing Characteristic Data of Agritourism Farms*

This paper presents the author's approach to the development of characteristic data of agritourism farms. The presented method makes it possible to identify objects with positive and negative assessments and to select a test object for further research as a case study. The research method was based on the transformation of features into a quantitative scale value. Despite the qualitative studies, the assessment results in scores for the following traits: negative (−2), medium negative (−1), standard (0), medium positive (+1) and positive (+2). By distributing the quantitative scores according to the given ranges, it is possible to keep the distances between the studied features equal, which improves the distinction between the features. To group farms with similar characteristics, it is necessary to apply the scheme described below:

- The farm (pn) assessed for the characteristics (Zmnx) of the I-VI category;
- A distance matrix was created from each feature (Zmnx) and added to the distance matrix of the respective category (I–VI);
- The distance matrix, for categories I–VI, is colored accordingly:

  (a)    Green—the nearest neighbor
  (b)    Red—the furthest neighbor

The tree diagram of the farm distance matrix illustrates the grouping of farms by similar characteristics. The units with the most similar characteristics were placed in one group. In contrast, the longer the distance between farms, the greater the difference between characteristics (Figure A3).

**5. Results and Discussion**

The region of Kłodzko (Ziemia Kłodzka) is very diverse and rich in cultural and natural heritage [9,58,59]. For a long time, the socio-economic, as well as cultural and natural conditions and numerous tourist attractions, have favored tourism and recreation in this area [4,9]. Drzewiecki [4] has drawn up a map showing the distribution of municipalities by the number of characteristics favourable to rural recreational space. He also assumed that municipalities meeting the criteria of rural agritourism space were those with at least three characteristics. According to this study, the district of Kłodzko is characterized by the so-called second degree of concentration of features beneficial to recreation. The second step corresponds between five and four identified beneficial traits. It is also important that since the 19th century, the villages of Ziemia Kłodzka (The Kłodzko Region) have been a popular place of recreation for the inhabitants of Silesian towns and cities [67]. The region has a very long tourist tradition due to its physiographic conditions [68]. Initially, it was a service for ramblers, summer visitors and health resort visitors providing accommodation in farmers' homesteads. At the end of the 19th century, following the expansion of roads, tourism has started to prevail in these areas. Due to uneconomic farming, the areas of Ziemia Kłodzka depopulated and entire villages transformed into summer resorts [67,68]. It is worth noting that the results of the survey on the preferences of visitors staying in holiday villages showed that respondents when choosing a village gave priority to its traditional character [9–13,64,67,69]. Research into agritourism also indicated that its development is influenced both by the so-called natural attractiveness factor (natural and landscape values) and cultural attractiveness (historical monuments, regional buildings) [7,9,60]. Therefore, it was considered that the area of Kłodzko District would be optimal for examining many indicators in terms of favoring agritourism as one of the economic sectors (tourism). The district of Kłodzko is particularly suitable for the provision of agritourism services due to historical and physiological conditions mentioned earlier. Research has shown that agritourism farms are an opportunity for rural development providing jobs or additional income for farmers. Such farms develop in many directions, creating separate areas of expertise. However, agritourism is also a threat of loss of mixed production (i.e., animal and plant production), and thus the cause of the decline in the diversity of agricultural production. As a result of the research, a method specifying how to approach the research

of agritourism farms was developed. It is a model approach to test the impact of particular features of a settlement unit on the cultural landscape.

Based on the changes recorded, a parametric assessment consisting of six categories was drawn up: homogeneity of the regional style (I), quality of the panoramas (II), environmental impact (III), local law (IV), the technical condition of building materials (V), agritourism space (VI). Agritourism farms were surveyed and assessed according to the characteristics assigned to each category (I–VI) (Table 1). Based on the analysis of quantitative data, results were obtained within these six categories of parameterization. First, characteristics common to all farms were analyzed to group farms with similar as well as divergent features. As a result of these analyses, groups of objects with similar characteristics within a given category were identified. Subsequently, a summary assessment of all categories was developed (Figure 6) to identify farms with the most and least desirable characteristics in terms of visual perception in the rural landscape of the region of Kłodzko.

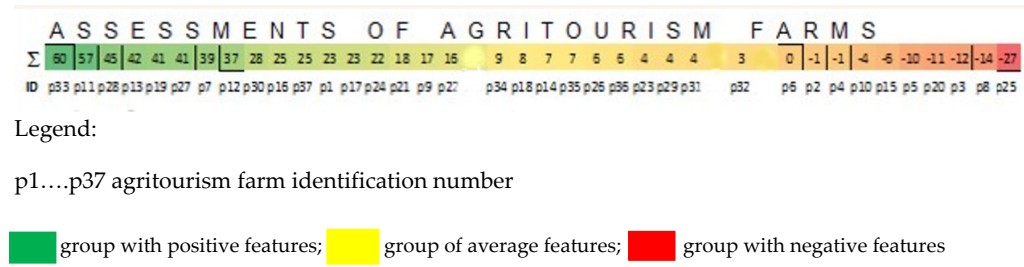

Legend:

p1….p37 agritourism farm identification number

green — group with positive features; yellow — group of average features; red — group with negative features

**Figure 6.** Distribution of aggregate assessment in the pool of surveyed agritourism farms.

Based on the summary assessment ranging from (−27) points to (+60) points, five farms with the highest negative score and six objects with the highest positive score were identified (Figure A4).

The analysis may help determine the guidelines and recommendations for shaping the agritourism space of the farm. The presented methodological/research approach for monitoring changes in the rural landscape can also help understand the current needs of people for different landscape services in rural areas. This is especially relevant in the context of sustainable landscape planning and management.

## 6. Conclusions

A pilot study of farms in the municipality of Stronie Śląskie began in 2015. That was the beginning of the field research and observations on agritourism farms of Ziemia Kłodzka. The proper research was completed in 2017 [62]. The collected data were then analyzed using the complete linkage clustering, which identified agritourism farms with similar characteristics. It is a model approach to test the impact of particular features of a settlement unit on the cultural landscape. The data collected through field trips to agritourism farms have been analyzed using qualitative and quantitative methods. The statistical study characterized research units in terms of location, area size, and type of agricultural production. The farms were then assessed according to the categories I–VI.

Based on a preliminary analysis of the statistical parameters, it was concluded that:

The research sample of agritourism farms is representative of all agritourism farms in the region of Kłodzko, which is confirmed by the fact that the distribution of values is close to a normal distribution;

- I, II and VI categories referring to the research sample in terms of their influence on the visual perception of the rural landscape in the region of Kłodzko, show the greatest variation;
- III, IV and V category do not make a significant difference to the characteristics of the test subjects in terms of the visual perception of the unit in the rural landscape; but they are important in terms of further development and environmental impact minimization;

- The owners of agritourism farms comply with the law, as they receive positive ratings in category IV;
- The study has indicated a correlation between the I and VI category, which confirms the differentiation of parametric values for these categories.

The above activities were aimed at selecting agritourism farms for the case study in further research (Figure 7). The results of analyses enable further research and help define guidelines and recommendations for future development of the agritourism space of the farm.

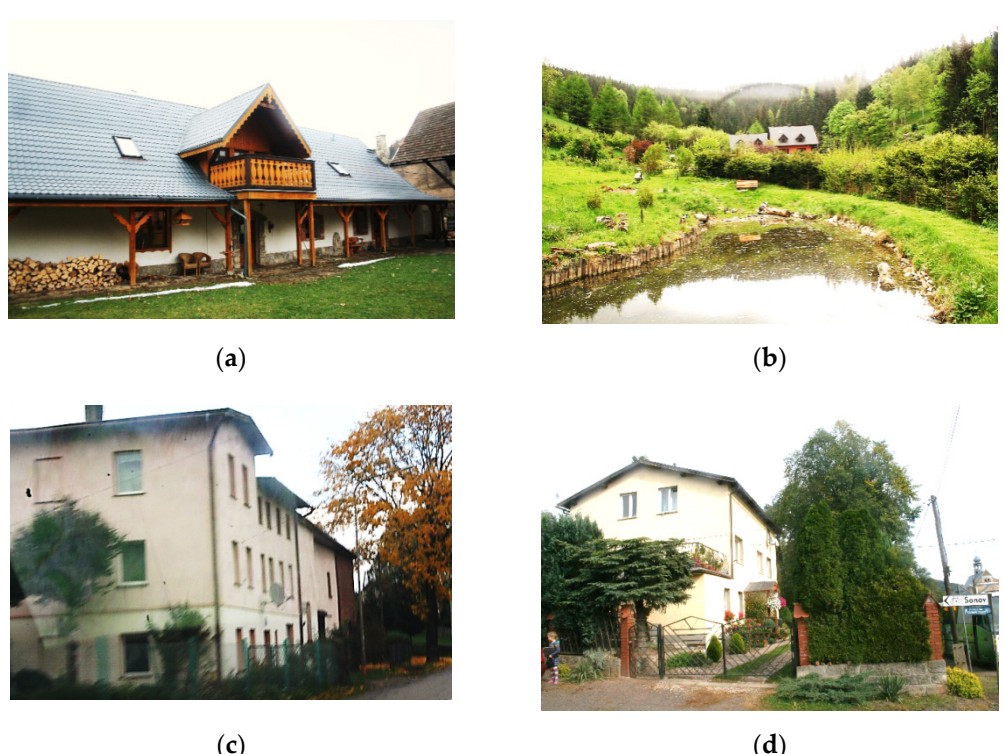

**Figure 7.** Agritourism (four examples out of eleven) selected for case study: (**a**) p33; (**b**) p13; (**c**) p25; (**d**) p3.

**Author Contributions:** Conceptualization, A.B.-S. and M.O.-D.; methodology, A.B.-S., M.O.-D. and W.J.; formal analysis, A.B.-S., M.O.-D. and W.J.; investigation, A.B.-S., M.O.-D., E.H. and W.J.; resources, A.B.-S., M.O.-D., E.H. and W.J.; data curation, A.B.-S.; writing—original draft preparation, A.B.-S.; visualization, A.B.-S. All authors have read and agreed to the published version of the manuscript.

**Funding:** This research received no external funding.

**Institutional Review Board Statement:** Not applicable.

**Informed Consent Statement:** Not applicable.

**Data Availability Statement:** Not applicable.

**Acknowledgments:** The study of agritourism farms was conducted as part of the doctoral dissertation of M.O.-D., supervisors E.H., A.B.-S. The authors would like to thank hosts of agritourism farms for their taking part in the survey. The authors would like to thank Martyna Lewandowska for their linguistic assistance.

**Conflicts of Interest:** The authors declare no conflict of interest.

## Appendix A

tree diagram

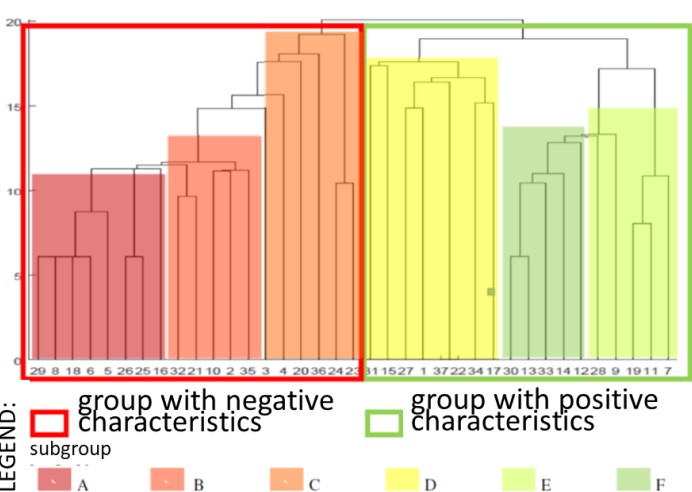

**Figure A1.** A hierarchical tree of parametric between grouped holdings with similar features.

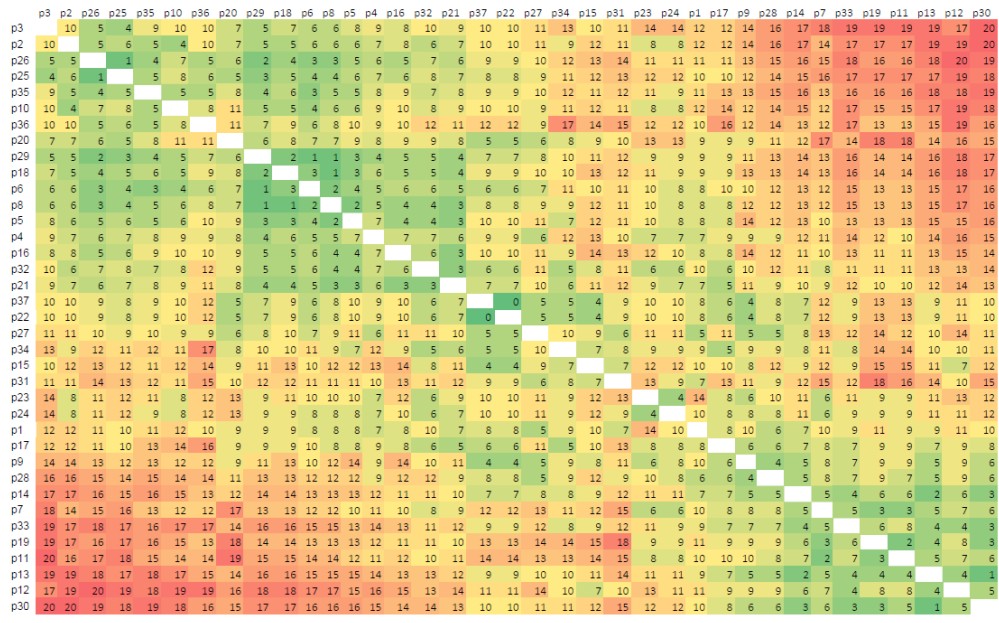

Legend:

p1....p37 agritourism farm identification number

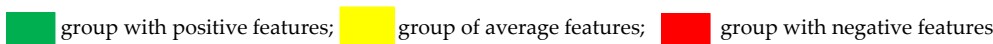

group with positive features;     group of average features;     group with negative features;

**Figure A2.** Example of a distance matrix evaluation for the category I The homogeneity of the regional style.

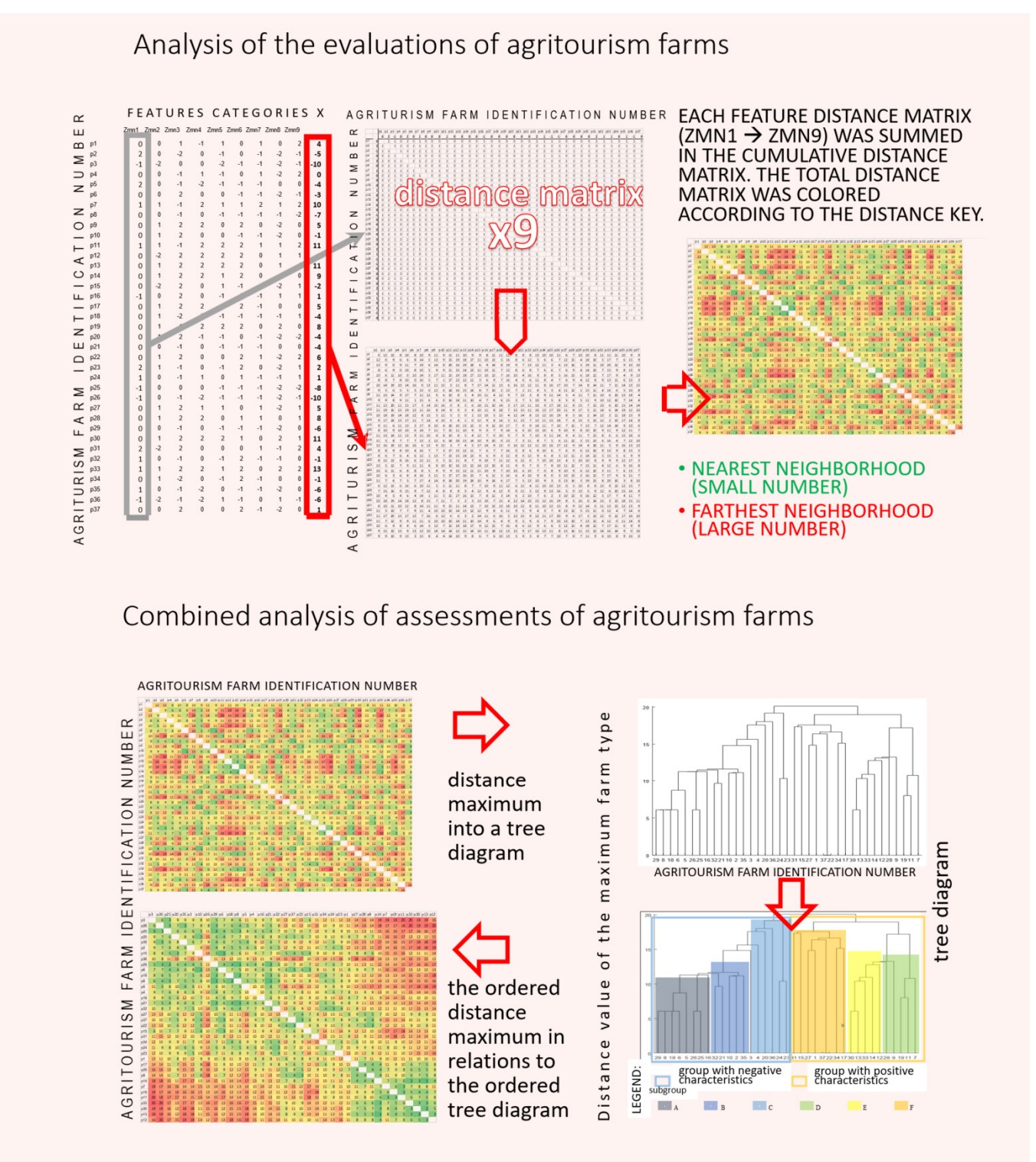

**Figure A3.** Scheme of transforming the distance matrix of one category into a hierarchical tree and ordered distance matrix in relation to the tree order.

| ID Agritourism farm | RATING | Remarks | STATUS | Category I The homogeneity of the regional style: | Category II The quality of the panoramas: | Category III Environmental impact: | Category IV Local law: | Category V Technical condition: | Category VI Agritourism space: greenery and details: |
|---|---|---|---|---|---|---|---|---|---|
| P33 | +60 | max.rating – cat.IV;max. rating cat.V, | ++ | all selected farms — the group with positive marks (group 2), all farms had buildings with main features of regional architecture, p33,p13 - same subgroup (F): roof windows were present - detail not in line with local architecture - but necessary in case of loft conversion into a residential function), p28,p11,p19 subgroup (E): different colours of elevations, roof coverings, p27 subgroup (D): regional detail was present). | p33,p11,p28,p19, p13 belong to the group (3) with positive marks and all these farms are characterised by high landscape values, p33,p11,p28,p19 are one of the subgroups( G ), in which greenery is taller than buildings, p13 is a subgroup (F), in which facilities were located near meadows, arable fields, and buildings were taller than greenery, p27 belongs to another group (2) with average characteristics (lower positive ratings, 0-2 points). | p33, p11, p13, p19, p27 - the group with positive marks (group 2, the same subgroup E), p28 belongs to the group with negative and neutral characteristics (group 1) to subgroup (B) with negative characteristics. This was caused by the fact that the farm used a solid fuel heating (the only one of the selected group for the case study with the highest positive ratings that had such heating). | p33,p28,p13,p19, p27 – the group with maximum positive marks (group 1, same subgroup A), development in line with the local plan, develops recreational activities and additional agritourism, whereas p11 belongs to the group 2 with neutral and negative ratings, subgroup (D) with medium (neutral) characteristics, where temporary space elements inconsistent with the local plan were present. | all selected farms — the group with positive marks (group 1), p33,p28 - subgroup (A) with the best technical condition of the facilities, p11, p13,p19, p27 belong to a subgroup (B) where the technical condition rating was at a good level. | all selected farms — the group with positive marks (group 3), p33, p11,p13, p27 same group (G), had additional educational and therapeutic space, but did not lose the space with a rural character, preserved the composition of the homestead greenery, pond.T-there were elements of cultural heritage (shrines, agricultural machinery), p28, p19 are a subgroup (F) with the highest positive ratings in this category, having space with elements of cultural heritage (e.g. pots, troughs, old equipment), residential, production and recreational zones, including a utility and ornamental garden with recreation; as well as diversity in terms of high greenery. |
| P11 | +57 | cat.V - a different group from the others (group 2 neutral and negative ratings) | ++ | | | | | | |
| P28 | +45 | max.rating –cat.IV max. rating cat.V, max.rating – cat.VI | ++ | | | | | | |
| P13 | +42 | max.rating –cat.IV | ++ | | | | | | |
| P19 | +41 | max.rating –cat.IV, max.rating –cat.VI, | ++ | | | | | | |
| P27 | +41 | max.rating –cat.IV | ++ | | | | | | |
| p5 | −10 | group 1 (subgroup A) the lowest score among the farms in cat.VI. | -- | all selected farms belong to the group with negative marks (group1), p5,p8,p25 are subgroup (A): brick buildings, with proportions that do not relate to regional buildings, p3 is a subgroup (B): dormer windows, slope windows, but correct roof slope) , p20 is a subgroup (C): some buildings on the holding with flat roofs, with different window openings in size and shape. | p5, p20,p8,p25 belong to the group with negative marks (group 1, subgroup C)These are p8,p20, p25 subgroup (A), p5 subgroup (B) ): represents the lowest scoring features, including the presence of low greenery, linear infrastructure within the habitat plot and low greenery around the plot (lawns alone), in terms of landscape values, the interiors of these habitats consist only of floor and background, and do not contain panorama components, p3 belongs to the group with medium marks (group 2), subgroup (D) rating 0-2. | p5, p8, p3, p25 belong to the group with negative marks (group 1, subgroup C)These are farms characterised by the use of gas for heating and water heating, the presence of hidden technical infrastructure and wasteland. All of them have sewage system and sort waste, they do not stand out due to the presence of materials protecting the environment against pollution (standard window and door woodwork, car park with loose soil, no insulation of the dwelling) , p20 subgroup (A) with the lowest ratings (solid fuel heating, septic tank, harmful materials, asbestos as roof covering). | all selected farms belong to the group of neutral and negative marks (group 2), p3, p8 (subgroup D ): buildings not in line with the local plan, building footprint exceeding 30% of the plot area), p20, p25 p5 (subgroup C): incompatibility of land use with use. | assessment in this category is very diverse, all belong to the group of neutral and negative marks (group 2), p3, p20 (subgroup E) buildings need renovation, moderately bad technical condition, p5, p8 (subgroup C) - good technical condition, p25 - (subgroup D) sufficient technical condition. | p3, p5, p8, p20, p25 belong to the group with negative marks (group 1), p5, p8, p20 - with the lowest score among the farms in this category: pseudo urban style dominates, small architecture made of artificial materials, mainly lawn, lack of diversified vegetation (subgroup A), p3, p25 (subgroup B): space is divided into residential, utility and recreational zones but mainly by fencing. |
| p20 | −11 | group 1, subgroup (A) with the lowest ratings (solid fuel heating, septic tank, harmful materials, asbestos as roof covering, group 1 (subgroup A) the lowest score among the farms in cat.VI. | -- | | | | | | |
| p3 | −12 | cat.II - a different group from the others (group 2 medium marks). | -- | | | | | | |
| p8 | −14 | group 1 (subgroup A) the lowest score among the farms in cat.VI. | -- | | | | | | |
| p25 | −27 | group 1 (subgroup A) the lowest scores among the farms in cat.I, group 1 (subgroup A) the lowest scores among the farms in cat.II. | -- | | | | | | |

**Figure A4.** List of agritourism farms selected for further research—case studies.

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
