# Peer review of "Parameterization in the Analysis of Changes in the Rural Landscape on the Example of Agritourism Farms in Kłodzko District (Poland)"

_sustainability, doi:10.3390/su14138026_

Round 1

Reviewer 1 Report

  1. The research objective of the paper is not very clear. Please add them in the introduction section.
  2. Figures 8-10 are not clear enough, the legend is too small, the authors should try to improve them.
  3. Methods and results should each be presented in separate sections, they should not be mixed.
  4. Interpretation and discussion of the results require further in-depth.
  5. The conclusion should be concise, so it needs further refined. Some of them can be moved into the result section.

Author Response

Dear Reviewer,
Thank you very much for reviewing our article. The comments and suggestions were very valuable to us.

Ad.1. The research objective is clearly stated in the introduction, as well as highlighted in the conclusion. The whole article has been reorganised and we have changed the format, we hope that it is now clearer.

Ad.2. Figures 8-10  have been corrected (figures and tables renumbered).

Ad.3.The article has been re-edited by us in this respect. The results have been presented separately from the methods.

Ad.4.We re-examined the literature in the context of our results.

Ad.5. We have improved the result section of the article to be concise. The entire article has been modified.

We have also improved parts of the text in terms of style and language. We have also added tables with additional, complementary information concerning our analysis, the study (tab.1, 4).

Yours sincerely,
Authors

Reviewer 2 Report

Dear authors,

Thank you for the opportunity of reviewing such an interesting manuscript. Authors provided the approach for monitoring changes in the rural landscape. The results confirm the relevance of the proposed methodology as an instrument for land management . All of these are particularly important for sustainable rural development.

The manuscript’s strengths. The general approach of the manuscript is especially good. The manuscript is informative and good structured. The title matches the content. The topic fits the Sustainability journal scope and the case is relevant. The introduction provide sufficient background and include sufficient references. Research methods were described exactly. The analysis has been performed reliably and the results have been presented in a clear manner. The conclusions match the research idea. Overall, the work deserves a high rating.

The manuscript’s weaknesses are given in the attached file.

Author Response

Dear Reviewer,

Thank you very much for your positive evaluation of our work and your valuable comments.

We have analysed and corrected all comments according to the pdf file. (our article).
To clarify, I will add that we did not correct the formula (bracket issue), it is about absolute values, so they must look like this (fig.7 now fig.6). We have checked the name Cluster Analysis, e.g. https://www.qualtrics.com/experience-management/research/cluster-analysis/ and some other sources, it is used.

We have also corrected some of the text for style and language. We also added tables with additional, complementary information about our analysis, study (tab.1, 4).

We have partially modified the article (form), we hope that it is now more readable (methods and results in order, separated).

Best Regards,
Authors

Reviewer 3 Report

While the English is sold, there are many poorly phrased sentences that do not make sense or are very hard to understand.

Literature review  - this is more a description of what a rural area is, rather than a systematic presentation on the theories or framework used in the paper. How is your research adding to this information? If you are measuring change, then your literature review should discuss the advantages or disadvantages of change. Some of this is covered in your introduction.

Methods – This seems to be a description of various methods, but not the methods actually used in this study. You should lay out exactly which methods you used and why, and then tell us where you collected data and how that data was analyzed. The goal is to provide a step-by-step ‘how to’ s that your research can be duplicated. As it stands, this is not a methods section.

Study area

I suggest moving the information about past studied into the literature review, which can be referred to in the study site section. These studies show the need for your research since they measure customer preferences, implying that change is a negative when relying on tourism.

Also, this section is a bit over detailed. I think the space could be better used for the literature review. You need to explain why you see the changes as a bad thing for tourism. It is obviously preferred by the farmsteads otherwise they would have chosen a different development model.

Research Approach

“The field research confirmed that there aren’t ten beds 287 in the majority of surveyed farms”. How many are there?

If figure 2 was discarded information, why is there a figure (which implies its importance)? We are more interested in what you did do, rather than what you did not do. This information could be described in a single sentence.

Please reference Table 1 in the text.

I am a bit confused about what is the basis for the parameterization. For example, what house color is considered appropriate (0), what color is +2 and what is -2? In other words, what is a positive feature and what is a negative feature (is adding a hot water heater considered negative)? I think this needs to be better explained for each of the variables so the reader knows how these are being graded and the criteria for ‘positive’ versus ‘negative’.

Conclusion

You state “The aim was to select outstanding 482 units from the research sample for further research as case studies”, yet in the introduction you claim that the goal is to “test the impact of particular features of a settlement unit on the 82 cultural landscape”, and in the introduction, you say” and in the abstract, to understand “the current needs of people for different landscape services 33 in rural areas”.

This section should reiterate what you found and tell us why it is important (by bring it back to your literature review). No new information should be put into the conclusion. Figure 7 should be in the methods section, not the conclusion.

Author Response

Dear Reviewer,

Thank you very much for reviewing our article. The comments and suggestions were very valuable to us.

We have corrected the entire article formally and added tables with additional information and data. We have also improved the whole article in terms of content. We have also improved parts of the text in terms of style and language.

We have mainly corrected the part of the article concerning the description of methods and results. The correction also concerned the conclusions.

 Parts of the article: Study area- have been shortened and information has been partly moved to other sections.

Research Approach -This entire section has been reworked and information has been added to clarify and complement our study (Tables 1,4).

The research objective is clearly stated in the introduction, as well as highlighted in the conclusion. The whole article has been reorganised and we have changed the format, we hope that it is now clearer.

The results have been presented separately from the methods.

Yours sincerely,

Authors

Round 2

Reviewer 1 Report

The title of the section 5 is missing?

Author Response

Dear Reviewer,
Thank you for your comments and resolution.
There was an error in the numbering of chapters and subchapters, it should be:

4.Methodological approach

4.1. Research trial;

4.2. Field research;

4.3. The extent of changes in the farm when changing its function from an agricultural to agritourism;

4.4. Assessment categories for agritourism farms;

4.5. Assessment Data Interpretation Method;

4.6.. Cluster analysis furthest neighbour (complete linkage), the maximum distance;

4.7. Methodology for analysing characteristic data of agritourism farms;

  1. Discussion and Results;
  2. Conclusions.

Best Regards,

Authors